# THE OVERCOOKED GENERALISATION CHALLENGE

## ABSTRACT

We introduce the Overcooked Generalisation Challenge (OGC) – the first benchmark to study reinforcement learning agents' zero-shot cooperation abilities when faced with novel partners *and* levels in the Overcooked-AI environment. This perspective starkly contrasts a large body of previous work that has evaluated cooperating agents only on the same level or with the same partner, thus failing to capture generalisation abilities essential for real-world human-AI cooperation. Our challenge interfaces with state-of-the-art dual curriculum design (DCD) methods to generate auto-curricula for training general agents in Overcooked. It is the first open-source cooperative multi-agent environment specially designed for DCD methods and, consequently, the first evaluated with state-of-the-art methods. It is fully GPU-accelerated, built on the DCD benchmark suite `minimax`, and freely available under an open-source license: `http://anonymised.edu`. We show that state-of-the-art DCD algorithms fail to produce useful policies on this novel challenge, even if combined with recent network architectures specifically designed for scalability and generalisability. As such, the OGC pushes the boundaries of real-world human-AI cooperation by enabling research on the impact of generalisation on cooperating agents.

## 1 INTRODUCTION

Developing computational agents capable of collaborating with humans has emerged as a key challenge in artificial intelligence (AI) research (Stone et al., 2010; Dafoe et al., 2020) and promises to vastly expand human abilities (O'neill et al., 2020). Recent years have seen considerable advances in understanding human cooperative behaviour (Rand & Nowak, 2013; Vizmathy et al., 2024), computational modelling of cooperation (Nikolaidis & Shah, 2013; Sadigh et al., 2016; Ding et al., 2024),as well as in developing computational methods for human-AI cooperation (Hu et al., 2020; Strouse et al., 2021). In parallel, several benchmarks (Samvelyan et al., 2019; Bard et al., 2020) were proposed to foster the development and evaluation of these methods. Most notably, Overcooked-AI (Carroll et al., 2019) has established itself as a widely used benchmark for evaluating (zero-shot) human-AI coordination (Strouse et al., 2021; Zhao et al., 2023; Yu et al., 2023).

Despite the advances they have enabled, all of these benchmarks are limited in that they only allow to assess reinforcement learning (RL) agents' cooperative abilities *in-distribution*. That is, they either only allow to evaluate agents in the same environment in which they were trained (Hu et al., 2020; Carroll et al., 2019) or with the same partner agent they were trained with (Foerster et al., 2018; Lowe et al., 2017; Strouse et al., 2021). In Overcooked-AI, for instance, existing zero-shot coordination (ZSC) methods are trained once per layout at considerable cost (Carroll et al., 2019; Yang et al., 2022; Zhao et al., 2023; Yu et al., 2023), and these layouts only feature a limited number of possible cooperation strategies (see Figure 1). However, real collaborative settings require coordination with novel partners in unknown environments. For example, consider a medical robot assisting doctors in hospitals. Such a robot will be deployed in unique and unknown hospitals and surgical rooms where they need to adapt to different medical staff and their preferences.

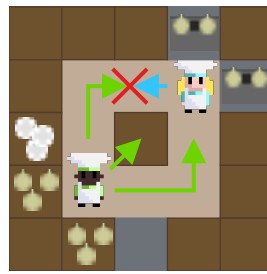

Figure 1: Coordination challenges in the Overcooked-AI *Coordination Ring* layout.

To address this limitation, we introduce the *Overcooked Generalisation Challenge* (OGC) – the first zero-shot cooperation benchmark that chal-

lenges agents to cooperate in novel layouts *and* with unknown partner agents. While previous open-source benchmarks studied opponents combined with map generalisation, no dedicated open-source benchmarks exist for studying cooperation partners combined with map generalisation. Cooperative settings differ from competitive ones in their game-theoretic background and thus require separate algorithms and benchmarks (Lerer & Peysakhovich, 2019). Neural MMO (Suarez et al., 2021; 2023) comes closest to our setting as it mixes cooperation and competition but crucially does not provide a purely cooperative setting which is specially designed for human-AI coordination – unlike Overcooked-AI. To train and evaluate agents on our benchmark, we make use of unsupervised environment design (UED) (Dennis et al., 2020) to generate suitable training levels, provide hand-designed testing levels, and asses zero-shot cooperation on these by providing populations of diverse testing agents. As such, our work is the first to combine UED techniques with a multi-agent RL zero-shot cooperation task and thus bridges the gap between two previously unrelated research areas; it studies the impact of generalisation on human-AI coordination and the ability of UED algorithms to design optimal auto-curricula for cooperating agents. We benchmark several UED algorithms and network architectures on our challenge and find that they struggle to perform well. Only PAIRED (Dennis et al., 2020), together with a policy that incorporates a soft Mixture-of-Experts (SoftMoE) module (Obando-Ceron et al., 2024), has some limited success at generalising to the testing levels and outperforms competitive baselines, including robust PLR (Jiang et al., 2021b;a) and AC-CEL (Parker-Holder et al., 2022). Overall, our findings call for developing methods that combine zero-shot coordination and DCD techniques in a single ZSC-DCD framework, and our benchmark provides the environment to do so. Taken together, our contribution is three-fold:

1. We introduce the Overcooked Generalisation Challenge – a novel benchmark challenge in which agents are asked to cooperate with novel partners in previously unseen layouts.

2. We provide OvercookedUED – an open-source environment that can be used with state-of-the-art DCD algorithms and that is integrated into `minimax` (Jiang et al., 2023), taking full advantage of the hardware acceleration provided by JAX.

3. We benchmark our environment by training agents with common DCD algorithms (Dennis et al., 2020; Jiang et al., 2021a; Parker-Holder et al., 2022) and show that current DCD algorithms struggle with the challenge even if we employ recent network architectures (Smith et al., 2023; Obando-Ceron et al., 2024). Furthermore, we assess zero-shot cooperation performance with a population of diverse partners to link zero-shot cooperation and generalisation. We show that as policies become more generally capable, they achieve better zero-shot cooperation.

## 2 RESEARCH CHALLENGES

The OGC poses several new challenges for zero-shot human-AI cooperation that go beyond existing benchmarks and that are essential for further advances in the development of cooperating RL agents:

**Generalisation**  The OGC challenges the generalisation capabilities of methods and agents by having them engage in a double generalisation challenge: adjusting to both novel partners and levels. Existing cooperative open-source benchmarks require typically only one form of generalisation, see for instance (Lowe et al., 2017; Foerster et al., 2018; Carroll et al., 2019; Hu et al., 2020).

**Environment Design**  Our environment challenges UED algorithms in generating and designing layouts with many interacting components and agents. This is in contrast to existing environments that only require UED algorithms to design simple mazes, 2D walker terrains, or race tracks consisting of fewer elements (Dennis et al., 2020; Jiang et al., 2021a; Parker-Holder et al., 2022; Rutherford et al., 2024a). We show that methods struggle to design layouts similar to the ones humans designed. Current methods specifically fail to design layouts requiring handing over items over a countertop or featuring deliberately designed circuits. Our benchmark challenges further research to develop UED methods that design more realistic collaboration environments for curriculum learning, possibly along the lines of Bruce et al. (2024).

**Combining Environment and Partner Generalisation**  Coordinating with novel partners and generalising to novel levels were often treated as separate research areas. As such population-based methods for zero-shot coordination do not apply to the challenge since training levels are generated

Table 1: Overview of benchmarks for unsupervised environment design and procedurally generated environments. Closed-source benchmarks are marked in gray – these cannot be evaluated on by the research community.

| Name | Multi-agent | Zero-shot coop. | GPU accel-erated | Open Source | Partial obs. | Img. obs. |
|---|---|---|---|---|---|---|
| XLand (Team et al., 2021; Bauer et al., 2023) | ✓ | ✓ | - | ✓ | ? | ✓ |
| LaserTag (Samvelyan et al., 2023) | ✓ | - | - | - | ✓ | ✓ |
| MultiCarRacing (Samvelyan et al., 2023) | ✓ | - | - | - | ✓ | ✓ |
| CoinRun (Cobbe et al., 2019) | - | - | - | ✓ | ✓ | ✓ |
| ProcGen (Cobbe et al., 2020) | - | - | - | ✓ | ✓ | ✓ |
| 2D Mazes (Cobbe et al., 2019; Dennis et al., 2020) | - | - | - | ✓ | ✓ | ✓ |
| CarRacing (Jiang et al., 2021a) | - | - | - | ✓ | ✓ | ✓ |
| Bipedal Walker (Wang et al., 2019) | - | - | - | ✓ | ✓ | - |
| AMaze (Jiang et al., 2023) | - | - | ✓ | ✓ | ✓ | ✓ |
| XLand-MiniGrid (Nikulin et al., 2023) | - | - | ✓ | ✓ | ✓ | ✓ |
| Craftax (Matthews et al., 2024) | - | ✓ | ✓ | ✓ | - | ✓ |
| JaxNav (Rutherford et al., 2024a) | ✓ | - | ✓ | ✓ | ✓ | - |
| **OvercookedUED (ours)** | ✓ | ✓ | ✓ | ✓ | ✓ | ✓ |

on the fly. Thus, training a best response against a diverse population on each layout is infeasible. There is currently no algorithm to train a population of diverse agents over a distribution of levels. Our benchmark encourages these branches to merge both lines of research and develop UED-ZSC methods, i.e. methods that learn both at the same time.

# 3 RELATED WORK

## 3.1 GENERALISATION IN REINFORCEMENT LEARNING

A large number of works have shown that RL agents fail to generalise to new environments, see (Zhang et al., 2018a; Cobbe et al., 2019), and have triggered research on the generalisation capabilities of RL agents (Nichol et al., 2018; Cobbe et al., 2019; 2020). Early results revealed that RL agents can memorise large numbers of levels during training (Zhang et al., 2018b; Cobbe et al., 2019) and that they must experience sufficiently diverse training data to generalise well (Cobbe et al., 2020). One established approach to generate diverse training data is to use domain randomisation (Jakobi, 1997, DR). Still, DR has been shown to produce many uninformative samples (Khirodkar et al., 2018), which can lead to the agent's inability to generalise (Dennis et al., 2020).

## 3.2 UNSUPERVISED ENVIRONMENT DESIGN

Intending to address this challenge, later works on generalisation focused on unsupervised environment design (Dennis et al., 2020, UED). UED aims to improve domain randomisation by generating auto-curricula that include training levels of increasing complexity to facilitate continued agent learning (Graves et al., 2017). It does so by adapting the free parameters of an under-specified environment to the agent's capabilities. Most popular UED methods fall into the category of Dual Curriculum Design (Jiang et al., 2021a, DCD) that combine 1) an agent, 2) a level generator, and 3) a curator that picks which levels to train on. Popular methods include Prioritised Level Replay (PLR) (Jiang et al., 2021b), robust PLR$^\perp$ (Jiang et al., 2021a), MAESTRO (Samvelyan et al., 2023), ReMiDi (Beukman et al., 2024), PAIRED (Dennis et al., 2020), ACCEL (Parker-Holder et al., 2022), and Replay-Enhanced (RE)PAIRED (Jiang et al., 2021a). While the development of these DCD methods has been steady, they have mostly been explored in simple environments, see Table 1.

**Single-agent UED Environments** Early work on generalisation mainly focused on single-agent environments (Zhang et al., 2018b; Farebrother et al., 2018; Cobbe et al., 2019) and these are also

popular in UED research. Among these, prior work has studied mazes (Dennis et al., 2020; Jiang et al., 2021a; Parker-Holder et al., 2022; Jiang et al., 2023; Li et al., 2023a; Beukman et al., 2024), bipedal walkers (Wang et al., 2019; 2020; Parker-Holder et al., 2022) or car racing environments (Jiang et al., 2021a). One likely reason for their popularity as benchmarks for DCD is that new levels are easy to generate, and agents are usually fast to train. However, they are limited to a single agent, with limited options to interact with the environment and other agents, and thus bear little resemblance to real-world problems.

**Multi-agent UED Environments** Compared to single-agent environments, multi-agent environments are inherently more complex because the agents interact with each other, as well as with the physical environment. Multi-agent environments are still rarely used in UED research. Most prominent is Deepmind's XLand (Team et al., 2021; Bauer et al., 2023), a closed-source multi-task universe for generating single- and multi-agent tasks and environments. While XLand features cooperative tasks, it is not available to researchers for studying cooperative multi-agent UED algorithms. While an open-source variant was recently published (Nikulin et al., 2023), it only supports a single agent. Arguably closest is Neural MMO (Suarez et al., 2021; 2023), which is a massively multi-task and multi-agent environment that mixes cooperation and competition to replicate massively multiplayer online games. We instead are interested in assessing and identifying cooperation performance in specially designed human-AI cooperation challenges for which the maissvely multi-task and multi-agent cooperation-competition setting of Neural MMO is unsuitable. Additionally, classic Overcooked already benefits from a rich history of human-AI cooperation research. Finally, while JaxNav (Rutherford et al., 2024a) features multi-agent path-finding no interaction between agents is required and the environment is not focused on human-AI cooperation. Other open-source environments are competitive, i.e. LaserTag (Lanctot et al., 2017; Samvelyan et al., 2023) and MultiCarRacing (Schwarting et al., 2021; Samvelyan et al., 2023), and thus not applicable to our setting. Opposed to all of these, our work contributes to and analyses the first open-source cooperative multi-agent UED environment.

### 3.3 Human-AI Cooperation in Overcooked-AI

Overcooked-AI (Carroll et al., 2019) has become one of the most important benchmarks for human-AI cooperation. The environment is fully cooperative and has two agents cook and deliver soups to earn a joint reward. Overcooked-AI was, for example, used in research on zero-shot cooperation (Strouse et al., 2021; Zhao et al., 2023; Yu et al., 2023; Li et al., 2023b; Yan et al., 2023, ZSC), language model-based cooperative agents (Liu et al., 2024; Tan et al., 2024), human modelling in cooperation (Yang et al., 2022). Zero-shot cooperation refers to cooperating with a partner not encountered during training. It is an important proxy to ensure the ability of an agent to coordinate with humans at test time, given that human data is often unavailable and agents thus must be able to coordinate effectively without previous training. It is commonly studied in Overcooked.

Related to our work is the work of Fontaine et al. (2021) in which the authors used procedurally generated Overcooked layouts to evaluate the impact of different layouts on human-robot interaction using planning algorithms. However, while they use procedural content generation in the Overcooked context their research does not focus on cross-layout generalisation – a major theme in our work. Our work is thus the first to explore the impact of cross-level generalisation for zero-shot cooperation and is the first to provide the necessary tools for this.

## 4 Preliminaries

The cooperative multi-agent UED setting can be formalised as a *decentralised under-specified partially observable Markov decision process* (Dec-UPOMDP) with shared rewards. A Dec-UPOMDP is defined as $\mathcal{M} = \langle \mathcal{N}, A, \Omega, \Theta, \mathcal{S}^{\mathcal{M}}, \mathcal{T}^{\mathcal{M}}, O^{\mathcal{M}}, \mathcal{R}^{\mathcal{M}}, \gamma \rangle$ in which $\mathcal{N}$ is the set of agents with cardinality $n$, $\Omega$ is a set of observations, and $\mathcal{S}^{\mathcal{M}}$ is the set of true states in the environment. Partial observations $o^i \in \Omega$ are obtained by agent $i \in \mathcal{N}$ using the observation function $O : \mathcal{S} \times \mathcal{N} \to \Omega$. Following Jiang et al. (2021a), a *level* $\mathcal{M}_\theta$ is defined as a fully-specified environment given some parameters $\theta \in \Theta$. In it, agents each pick an action $a_i \in A$ simultaneously to produce a joint action $\boldsymbol{a} = (a_1, \ldots, a_n)$ and observe a shared immediate reward $R(s, \boldsymbol{a})$. Then, the environment transitions to the next state according to a transition function $\mathcal{T} : \mathcal{S} \times \mathcal{A}^1 \times \ldots \times \mathcal{A}^n \times \Theta \to \Delta(\mathcal{S})$ where

Figure 2: Overview of the Overcooked Generalisation Challenge (OGC) and how it is typically used in a Dual Curriculum Design (DCD) algorithm. The OGC supports teacher-based methods like PAIRED (Dennis et al., 2020) via unsupervised environment design (UED) and edit-based methods like ACCEL (Parker-Holder et al., 2022) via mutator functions of existing layouts.

$\Delta(\mathcal{S})$ refers to the space of distributions over $\mathcal{S}$. $\gamma \in [0, 1)$ specifies the discount factor. Agents learn a policy $\pi$. The joint policy $\boldsymbol{\pi}$ together with the discounted return $R_t = \sum_{i=0}^{\infty} \gamma^i r_{t+1}$ induce a joint action value function $Q^{\boldsymbol{\pi}} = \mathbb{E}_{s_{t+1:\infty}, \boldsymbol{a}_{t+1:\infty}}[R_t | s_t, \boldsymbol{a}_t]$. The definition of the Dec-UPOMDP extends a Dec-POMDP (Oliehoek & Amato, 2016; Wu et al., 2021) with the free parameters of the environment $\Theta$, analogously to previous works (Dennis et al., 2020; Jiang et al., 2021a; Samvelyan et al., 2023). Our definition differs from (Samvelyan et al., 2023) in terms of the shared rewards and general-sum nature. Within our Dec-UPOMDP, we perform UED to train a policy over a distribution of fully specified environments that enable optimal learning. This is facilitated by obtaining an *environment policy* $\Lambda$ (Dennis et al., 2020) that specifies a sequence of environment parameters $\Theta^T$ for the given policy that is to be trained. How $\Lambda$ is obtained depends on the DCD method. For example, in OvercookedUED, $\Theta$ represents the possible positions of walls, pots, serving spots, agent starting locations, and onion and bowl piles which is adjusted by $\Lambda$ throughout training.

## 5 THE OVERCOOKED GENERALISATION CHALLENGE

An overview of the *Overcooked Generalisation Challenge* is shown in Figure 2. The OGC extends previous work by evaluating the cooperative abilities *out-of-distribution*. That is, in contrast to existing UED environments, the OGC focuses on the cooperation of multiple agents in a complex, cooperative task across different levels. More specifically, two different agents are tasked with cooking a soup together in the five original layouts of Overcooked-AI (see Figure 3), but without having encountered them and their partner during training. Since the original five layouts have been designed to test and explore different kinds of cooperation, they form suitable out-of-distribution test levels. To train an agent capable of generalisation, we generate a curriculum of possibly endless diverse training layouts via procedural content generation. The OGC is more closely aligned with real-world human-AI collaboration as it does not limit evaluation to one specific physical environment or partner. To generate a curriculum of layouts, we use DCD methods. Specifically, methods in which an environment designer interacts with the challenge by designing layouts either from scratch through interacting with *OvercookedUED* – a novel environment for creating Overcooked levels – by alternating existing layouts through the *Overcooked mutator* or by letting the OGC generate random layouts. At every step of the curriculum, this designer must account for agents' cooperation ability when trying to generate layouts that are at the forefront of their abilities.

### 5.1 COMPONENTS OF THE CHALLENGE

While OGC refers to the challenge as a whole, it comprises several components that enable its integration with DCD algorithms (see Figure 2). At the heart of it, it features an Overcooked environment capable of running different levels fast and in parallel in which agents learn to cooperate. It features *OvercookedUED* that provide methods, interfaces and a teacher environment

Figure 3: We study the five layouts proposed by Carroll et al. (2019). From left to right: *Cramped Room*, *Asymmetric Advantages*, *Coordination Ring*, *Forced Coordination*, and *Counter Circuit*.

to design novel layouts as well as an *Overcooked Mutator* that alters existing layouts, specifically designed to be used with ACCEL.

**Overcooked-AI**  OGC builds on the Overcooked-AI environment. We adapted the version provided by the JaxMARL project (Rutherford et al., 2024b), keeping features consistent with the original implementation. This includes action and observation spaces, i.e. the set of actions is {left, right, up, down, interact, stay} and observations are encoded as a stack of 26 $h \times w$ boolean masks encoding the positions of elements in the environment. In this representation, the first mask encodes the position of the first agent, the second mask the one of the second agent etc. Since agents now learn to play on many different layouts all at once, we adjust the environment to be capable of parallelising across differently shaped levels via padding. I.e., during rollouts, layouts are padded to a maximum size, and all objects in these layouts are one-hot encoded based on their position in equally sized masks. While this facilitates fast parallel rollouts that can be just-in-time compiled, it requires the introduction of a maximum height $h$ and width $w$ that need to be picked as a hyperparameter before training.

**OvercookedUED**  OvercookedUED features the interfaces necessary to design new layouts. For algorithms that make use of a teacher agent to create layouts (PAIRED, etc.), OvercookedUED provides a teacher environment (see Figure 2). This environment allows a teacher policy to take design steps to parameterise the underspecified MDP. At every timestep $t$ of the generation process the teacher observes the unfinished layout and picks an action from a space that consists of the total number of cells in the $h \times w$ grid. This cell then becomes filled with the next items to be placed. Objects are placed sequentially and in a deterministic order, starting from walls, agents one and two, goal, onion, pot and bowl positions. An episode in the teacher MDP lasts until all items are placed. In case of a conflict, elements are placed randomly on free cells. The teacher is parameterised by its own neural network. As in previous work (Jiang et al., 2023), OvercookedUED does not check whether a layout is solvable and leaves the task of designing and/or identifying suitable training layouts to the DCD method.

For algorithms that do not specify a teacher, such as PLR, OvercookedUED generates random layouts. These random layouts feature one or two piles of onions, bowls, pots and serving locations, and two agents.

Finally, some DCD algorithms, such as ACCEL, require alternating existing layouts by mutating them. OvercookedUED supports layout mutation through five basic operations: (1) converting a random wall to a free space and vice versa, (2) moving goals, (3) pots, (4) plate piles, and (5) onion piles. Given a layout, our *mutator* randomly samples $n$ operations and applies them. All versions allow the number of walls placed to be configured and the environment always places a border wall.

**Implementation**  The OGC is implemented in Jax (Bradbury et al., 2018) and integrated into minimax (Jiang et al., 2023). As such, it can be tested with all available DCD algorithms present in minimax. To achieve this we extend minimax with runners, replay buffers etc. that are compatible with multiple agents. Building on an established library eliminates sources of error and presents users of the challenge with a familiar experience. We present the steps-per-seconds (SPS) on our setup given varying degrees of parallelism in Table 2 and compare it to the GPU-accelerated maze environment minimax includes AMaze. Given sufficiently large numbers of parallel environments, OGC can be run at hundreds of thousands of SPS. While less than AMaze, the OGC is a more fully-featured environment in which multiple agents take steps and interact.

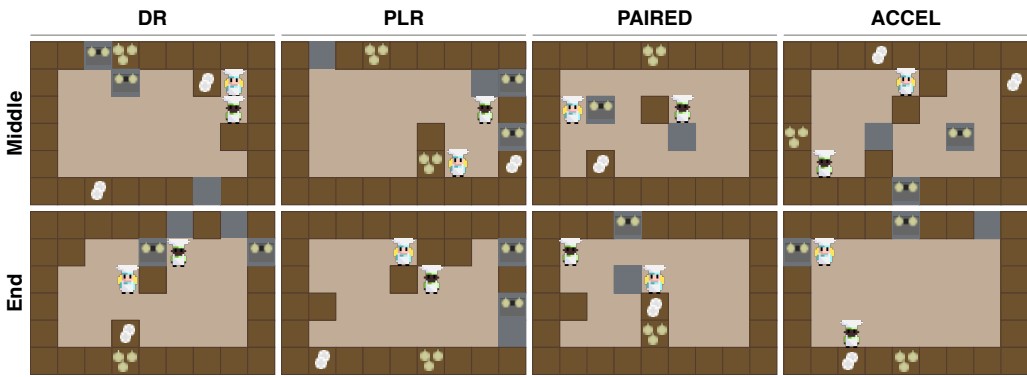

Figure 4: Sample levels generated by the different methods after $15,000$ (Middle) and $30,000$ (End) epochs. Even after considerable training, none of the methods can guarantee the generation of solvable layouts (Middle-row leftmost and rightmost).

## 5.2 EVALUATION

We evaluate agents by their performance on out-of-distribution Overcooked-AI layouts to asses generalisation performance in self-play and in cross-play. In cross-play, a fictitious co-play (Strouse et al., 2021, FCP) and maximum entropy-based population based training (Zhao et al., 2023, MEP) population of a total 24 agents each is used to asses zero-shot cooperation. Both

Table 2: Average steps-per-second for different numbers of parallel environments measured by taking 1,000 steps with randomly sampled actions.

| # Parallel Envs | 1 | 32 | 256 | 1024 |
|---|---|---|---|---|
| AMaze | 264 | $8,141$ | $67,282$ | $264,142$ |
| OvercookedUED | 151 | $4,921$ | $40,011$ | $156,696$ |

populations are trained with equal settings and include a low, medium and high-skilled checkpoint of each run extracted at 10, 50 and 100 % achieved return respectively. The population entropy coefficient $\alpha$ is 0.01 for MEP. In this work we define zero-shot coordination as the task of cooperating with a partner, which has not been encountered during training and view it in contrast to ad-hoc teamwork (Stone et al., 2010) since in our setting there is no time to update a fixed policy after training (Hu et al., 2020). As zero-shot cooperation with a diverse population has become a proxy for assessing the abilities of an agent to coordinate with humans. Our benchmark includes the necessary tools to perform this evaluation. In our analysis, we report results using the mean episode reward and mean layout solved rate, similar to previous work (Jiang et al., 2023). A layout is considered solved if an agent pair delivers more than one soup which differentiates goal-directed from random behaviour. We present these metrics in the self- and cross-play settings. Additionally, we investigate what kinds of levels agents perform poorly in and why in a final error-analysis.

## 6 ANALYSING & BENCHMARKING THE CHALLENGE

We benchmark the challenge with several DCD algorithms and network architectures. We aim to set a performance baseline for future works and show what evaluations this benchmark enables. To this end, we first show that generalising to novel layouts in Overcooked is difficult, and then we move on to the additional challenge of zero-shot cooperation.

All baselines are trained using MAPPO, which is known to work well in cooperative settings (Yu et al., 2022) using centralised training and decentralised execution (Foerster et al., 2016). As for DCD algorithms, we compare the performance of DR, $\text{PLR}^{\perp,\parallel}$, Pop. PAIRED and $\text{ACCEL}^{\parallel}$. We chose these methods as they have better theoretical guarantees ($\text{PLR}^{\perp}$ vs PLR), better runtime performance ($\text{ACCEL}^{\parallel}$ and $\text{PLR}^{\parallel}$), or because we found them to perform better empirically (Pop. PAIRED vs PAIRED). We excluded POET (Wang et al., 2019) in this analysis as it outputs specialists rather than generalists, which we require (Parker-Holder et al., 2022). Additionally, we excluded

Table 3: Mean episode reward for the different methods averaged over the respective testing layouts. The best result is shown in **bold**. We report aggregate statistics over three random seeds. We include Oracles which were trained on the five testing layouts directly to establish an empirical maximum.

| Method | CNN-LSTM | SoftMoE-LSTM | CNN-S5 |
|---|---|---|---|
| DR | $0.46 \pm 0.16$ | $5.22 \pm 7.19$ | $0.00 \pm 0.00$ |
| PLR$^{\perp,\parallel}$ | $0.17 \pm 0.06$ | $0.91 \pm 0.71$ | $0.12 \pm 0.15$ |
| Pop. PAIRED | $0.19 \pm 0.09$ | $\mathbf{13.34 \pm 5.70}$ | $0.24 \pm 0.19$ |
| ACCEL$^{\parallel}$ | $0.20 \pm 0.14$ | $0.67 \pm 0.60$ | $0.28 \pm 0.26$ |
| Oracle | $\mathbf{189.49} \pm 12.96$ | $\mathbf{217.02} \pm 39.18$ | $\mathbf{155.01} \pm 12.82$ |

MAESTRO as it is based on prioritised fictitious self-play (Heinrich et al., 2015; Vinyals et al., 2019) that is not easily adaptable to the cooperative setting (Strouse et al., 2021). As in (Jiang et al., 2023), if not stated otherwise, we train in 32 parallel environments and stop after $30,000$ outer training loops, amounting to just under 400 million steps in the environment. Hyperparameters were picked after a grid search over reasonable values, and all parameters are provided in Appendix A.4. Our default neural network architecture consists of a convolutional encoder with a recurrent neural network with an LSTM (Hochreiter & Schmidhuber, 1997). It is picked for its good performance in previous work (Yu et al., 2023) (see Appendix A.5 for details). In addition to our default network architecture, we explore the use of SoftMoE (Obando-Ceron et al., 2024), which have recently been identified for their potential for enabling scaling and generalisation, and S5 layers (Smith et al., 2023) due to the strong results of structured state-space models (Gu et al., 2022) in meta reinforcement learning (Lu et al., 2023). SoftMoE modules replace the penultimate layer after the feature extractor and S5 layers the LSTM in all experiments. We hypothesise that these provide better generalisation performance. Using these parameters, we verified that agents also overfit to their level in Overcooked by evaluating agents trained on a single layout on all layouts (cf. Appendix A.6.1). Additionally, we verified that all architectures can be fitted to the testing layouts when trained on them directly. We will refer to these as *Oracles* and use them to establish the maximum performance possible. Lastly, for all runs we display training curves on the seen and the five unseen evaluation levels in Appendix A.7.1.

**Layout Generalisation Performance** Simply generalising to the testing layouts in the OGC is already challenging for all methods without having to coordinate with novel partners, as presented in Table 3. Compared to commonly used single-agent Maze environments (such as AMaze, compare (Jiang et al., 2023)), all DCD methods struggle to obtain good results. This is most evident when compared with oracle policies (bottom row). PAIRED outperforms all other models significantly $0.01 < p < 0.05$ using a one-sided paired t-test. This is also shown in the mean solved rate where it reaches $14.6 \pm 7.7\%$, while all other models have a solved rate of mostly $0\%$ (cf. Appendix A.6.2). While this model performs better on average, layouts differ greatly in their difficulty. Our best-performing model reaches modest performance in Asymmetric Advantages and Cramped Room while mostly failing in the others, with no other model achieving noteworthy results. Recall that the environment features more moving parts that must be placed correctly to facilitate learning. This makes it hard for approaches like DR to find optimal placements by pure chance, as reflected in the results. The full results are in the Appendix A.6.3.

**Zero-Shot Cooperation Performance** Ultimately, we want the OGC to connect map generalisation and zero-shot coordination. To that end, we train and then use a FCP and MEP population (see Appendix A.6.4 for details) to establish how general cooperative agents can coordinate with diverse policies. We present preliminary results in Figure 5 together with two other baselines: *stay* which is a partner that never moves and *random* which samples random actions. As performance on out-of-distribution levels rises, agents become more competent at zero-shot cooperation. PAIRED always outperforms baselines (cf. Appendix A.6.5). However, even PAIRED policies often perform only slightly better than random baselines, which signifies the challenges of our benchmark. This is also evidenced by the kinds of levels these methods generate (Figure 4), as they tend to pivot towards generating open spaces that ease cooperation but are notably different from evaluation layouts. Overall,

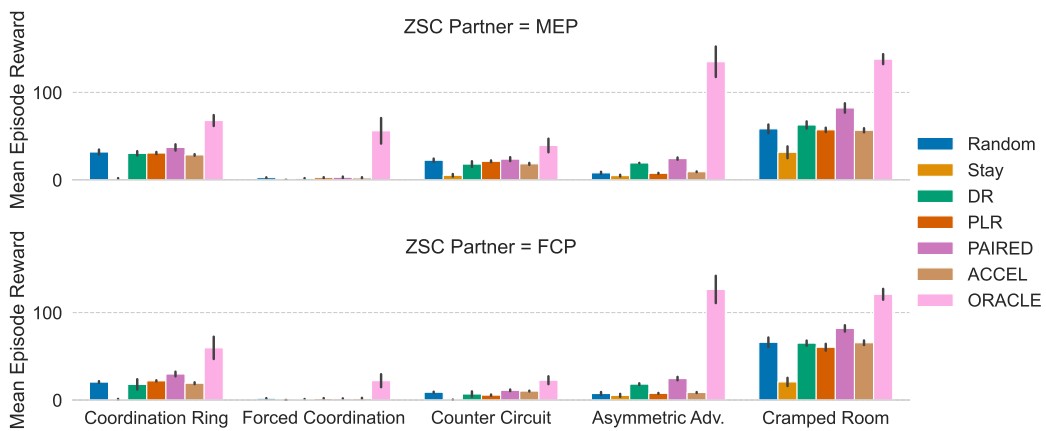

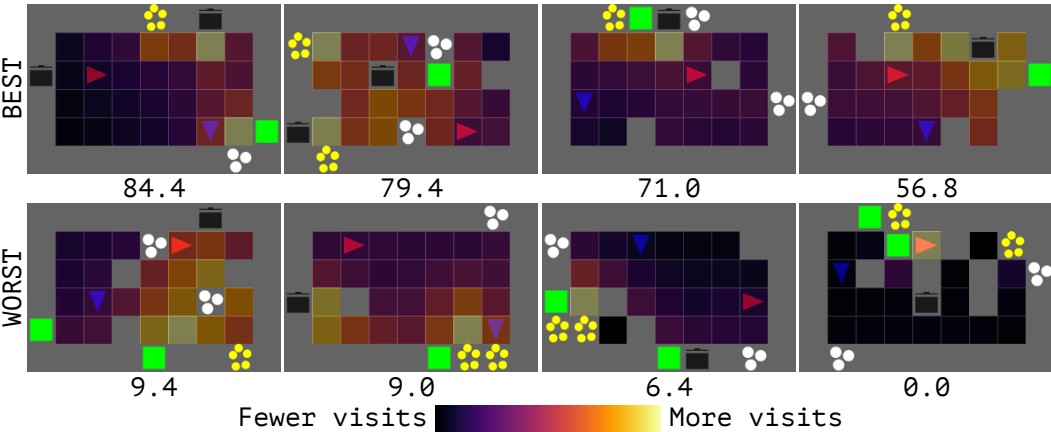

Figure 5: Zero-shot coordination results of the SoftMoE-LSTM policy paired with an FCP population trained on the respective layout. We report the mean episode reward and standard error.

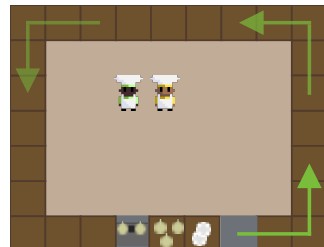

Figure 7: Sample levels that our models perform best (top) and worst in (bottom). The number of visits to each grid cell is shown as a heatmap overlay, while the mean return is stated below each layout. The layouts, where the model performs worst in tend to feature narrow elements or large distances between items.

cooperation performance is mostly carried by the expert FCP and MEP agents (compare Tables 13 and 14), mostly since our agents struggle to perform on the evaluation layouts in the first place.

**Error Analysis** We perform two final experiments to investigate the poor performance of our baselines and to eliminate trivial sources of errors. First, we hypothesise that the top-down observations in OGC are hard to generalise from since they are not invariant to mirroring or rotations (Ye et al., 2020). To test this we evaluate agents on 24 hand-designed circular evaluation levels with different kinds of symmetry, as shown in Figure 6. We find that agents tend to perform similarly across these layouts as the standard deviation is at most 1.1, and therefore reject this hypothesis (more details in Appendix A.7). Second, we investigate the kinds of levels our best-performing model does well vs poorly in from a pool of randomly generated evaluation levels in Figure 7. The figure summarises the cooperation behaviour of the agents by showing which cells are visited most frequently to give an impression of their motion patterns. While on many layouts our model reaches good self-play performance (up to a maximum mean reward of 84.4; top row), it typically

Figure 6: An illustration of the circular evaluation levels; we move the kitchen around the sides and vary the size.

delivers few to no soups in layouts it performs worst in. These levels tend to be narrow/convoluted and/or feature big distances between objects. Notice that the training levels in which our model performs well in are similar to Asymmetric Advantages and Cramped Room, while the worst levels are similar to the other 3 evaluation levels. In conclusion, current DCD methods struggle with generating training layouts of the correct complexity, i.e. ones that are similarly hard to evaluation ones.

**Discussion** Previous work (Jiang et al., 2021a) has found that PLR$^\perp$ tends to outperform the other here-tested algorithms in navigation-based tasks. Our more challenging environment suggests that this might not always be the case. In our preliminary analysis, PAIRED outperformed other DCD methods. Compared to mazes, car racing, or walker environments with fewer moving pieces, Overcooked layouts are more complex to design, requiring the designer to place multiple objects in relation to each other and the agents. Methods that employ a random generator therefore struggle in such a big design space. This thus requires a capable generator and suggests that simple navigation-based environments used to benchmark DCD in UED algorithms do not allow full performance evaluation. As such, OvercookedUED can be an important part of evaluating DCD algorithms. We envision that general Overcooked agents should be evaluated in scenarios that are difficult for self-play agents using our benchmark. These include zero-shot cooperation with strongly-biased agents (Yu et al., 2023) in Coordination Ring (see Section 1) and Asymmetric Advantages as described in (Ruhdorfer, 2023) and for which we provide the tools.

## 7 LIMITATIONS

Despite its many advantages, our challenge has two limitations. First, we artificially restricted the maximum size of the layouts to allow the environment to be both fully observable as in Carroll et al. (2019) and parseable by CNN-based feature encoders. Future work should focus on more natural representations of the whole scene, e.g. using graphs or item embeddings. While we included a partial observation that could theoretically be computed independently of size, similar to the vector-based observation used for behaviour cloning agents in (Carroll et al., 2019), batching across layouts in OvercookedUED still requires the layouts to be scaled to the same height and width. Second, while our challenge allows us to study zero-shot coordination via generalising across layouts, reasoning about other agents (Rabinowitz et al., 2018; Gandhi et al., 2021; Bara et al., 2023; Bortoletto et al., 2024b;a) might be equally important to achieve zero-shot cooperation capabilities on unknown layouts. This is plausible given that humans can reason about the mental states of other agents via Theory of Mind (Premack & Woodruff, 1978), as well as the physical configuration of the space in which they operate. Future work could thus explore reasoning about other agents in previously unexplored environments.

## 8 CONCLUSION

We have presented the Overcooked Generalisation Challenge (OGC) – a generalisation challenge focusing on (zero-shot) cooperation in MARL in out-of-distribution test levels. Our challenge is the first open-source cooperative multi-agent UED environment and is significantly more challenging than previous environments commonly used in UED and DCD research. In addition to using the challenge in UED research, we have shown how the OGC can be used in future research on human-AI collaboration as a zero-shot cooperation benchmark for general agents. That is, our challenge establishes a link between generalisation and zero-shot coordination. Our work is the first to provide the research community with the tools to train and evaluate agents capable of coordinating in previously unknown physical spaces and with novel partners.

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

# A APPENDIX

## A.1 Accessibility of the benchmark

We make our challenge available under the Apache License 2.0 via a code repository: `https://anonymised.edu`. Our environment is built on top of the existing `minimax` project (accessible under Apache License 2.0 via `https://github.com/facebookresearch/minimax`) and is thus accessible to researchers who are already familiar with the project. `minimax` is extensively documented, fast, and supports multi-device training. For all details, including a full description of the advantages of `minimax`, we kindly refer the reader to the accompanying publication (Jiang et al., 2023). Our Overcooked adaption is extended from the one in JaxMARL also accessible under Apache License 2.0 via `https://github.com/FLAIROx/JaxMARL`. Our code includes extensive documentation and examples of how it may be used. Additionally, our code is written in a modular fashion and other multi-agent environments can be integrated with the runners thanks to the careful design of the original project.

## A.2 Broader impacts

While our work is largely foundational and concerned with providing the research community with the appropriate tools for the training and evaluation of agents in game-like environments, special caution is always imperative should this research be applied to human-AI collaboration. Even though our goal is to improve collaboration, safeguards should be applied to make sure that humans are always safe from harm. Especially so in real-world applications where accidents could potentially result in bodily harm. Since our work is still far removed from any real-world application, we do not expect that our work in its present form carries the risk of materialising these harms. Some form of unsupervised environment design in collaborative environments might be part of future systems and we therefore acknowledge these risks. This work of course also carries the potential to improve human-AI collaboration and we make an important contribution to advancing the field with potential impacts in all kinds of human-machine interaction.

## A.3 Infrastructure & tools

We ran our experiments on a server running Ubuntu 22.04, equipped with NVIDIA Tesla V100-SXM2 GPUs with 32GB of memory and Intel Xeon Platinum 8260 CPUs. All training runs are executed on a single GPU only. We trained our models using Jax (Bradbury et al., 2018) and Flax (Heek et al., 2023) with 1, 2 and 3 as random seed for training DCD methods and 1 to 8 as random seeds for the populations. Training the DCD methods usually finishes in under 24 hours, only SoftMoE and PAIRED-based methods take longer. SoftMoE-based policies often take an extra 50% wall-clock time to train. Noticeable is also that our S5 implementation is the fastest, usually needing 30% less time. Both are compared to the default architectures' training time. In the longest case, the combination of a SoftMoE-LSTM policy trained with PAIRED takes about 80 hours to complete training. Our benchmark should be runnable on any system that features a single CUDA-compatible GPU. Although in our experience our experiments will require 32GB VRAM to run.

## A.4 Hyperparameters

We overview all hyperparameters for training in Table 4 and provide details on the hyperparameter search used in Table 5. This search was conducted on smaller single layout runs to determine reasonable values as complete runs would have been computationally infeasible. Furthermore we show the hyperparameters for each DCD method separately: DR hyperparameters in Table 6, PLR hyperparameters in Table 7, ACCEL hyperparameters in Table 8, and PAIRED hyperparameters in Table 9. DR hyperparameters govern how Overcooked levels are generated randomly and apply to all other processes in which a random level is sampled, for instance, in PLR, in which case the same hyperparameters apply.

## A.5 Neural network architectures

This work employs an actor-critic architecture using a separate actor and critic in which the critic is centralised for training via MAPPO (Yu et al., 2022). For the actor, the observations are of shape $h \times w \times 26$, while for the centralised critic, we concatenate the observations along the last axis to

Table 4: Hyperparamters of the learning process.

| Description | Value |
| --- | --- |
| Optimizer | Adam (Kingma & Ba, 2015) |
| Adam $\beta_1$ | 0.9 |
| Adam $\beta_2$ | 0.999 |
| Adam $\epsilon$ | $1 \cdot 10^{-5}$ |
| Learning Rate $\eta$ | $3 \cdot 10^{-4}$ |
| Learning Rate Annealing | - |
| Max Grad Norm | 0.5 |
| Discount Rate $\gamma$ | 0.999 |
| GAE $\lambda$ | 0.98 |
| Entropy Coefficient | 0.01 |
| Value Loss Coefficient | 0.5 |
| # PPO Epochs | 8 |
| # PPO Minibatches | 4 |
| # PPO Steps | 400 |
| PPO Value Loss | Clipped |
| PPO Value Loss Clip Value | 0.2 |
| Reward Shaping | Yes (linearly decreased over training) |

Table 5: Values used for a grid search over hyperparameters governing the learning process. Finally used values appear in **bold**.

| Description | Value |
| --- | --- |
| Learning Rate $\eta$ | $[1 \cdot 10^{-4}, \mathbf{3 \cdot 10^{-4}}, 5 \cdot 10^{-4}, 1 \cdot 10^{-3}]$ |
| Entropy Coefficient | $[\mathbf{0.01}\ 0.1]$ |
| # PPO Steps | $[256, \mathbf{400}]$ |
| # Hidden Layers | $[2, \mathbf{3}, 4]$ |
| Reward Shaping Annealing Steps | $[0, 2500000, 5000000, \textbf{until end}]$ |

form a centralised observation, i.e. the centralised observation has shape $h \times w \times 52$ following prior work (Yu et al., 2023).

All our networks feature a convolutional encoder $f_c$. This encoder always features three 2D convolutions of 32, 64 and 32 channels with kernel size $3 \times 3$ each and pads the input with zeros. Our default activation function is ReLU (Fukushima, 1975; Nair & Hinton, 2010) which we apply after every convolutional block. We feed the output of $f_c$ to a feed-forward neural network $f_e$ with three layers with 64 neurons, ReLU and LayerNorm (Ba et al., 2016) applied each. $f_e$ takes the flattened representation produced by $f_c$ and produces an embedding $e \in \mathbb{R}^{b \times t \times 64}$ that we feed into a recurrent neural network (either LSTM (Hochreiter & Schmidhuber, 1997) or S5 (Smith et al., 2023)) to aggregate information along the temporal axis. We use this resulting embedding $e_t \in \mathbb{R}^{b \times 64}$ to produce action logits $l \in \mathbb{R}^{b \times 6}$ to parameterise a categorical distribution in the actor-network or directly produce a value $v \in \mathbb{R}^{b \times 1}$ in the critic network using a final projection layer. This architecture is inspired by previous work on Overcooked-AI, specifically (Yu et al., 2023), see Figure 8 for an overview. We also test the use of a S5 layer (Smith et al., 2023) in which case we use 2 S5 blocks, 2 S5 layers, use LayerNorm before the SSM block and the activation function described in the original work, i.e. $a(x) = \text{GELU}(x) \odot \sigma(W * \text{GELU}(x))$.

In the case of the SoftMoE architecture, we follow the same approach as in (Obando-Ceron et al., 2024) and replace the penultimate layer with a SoftMoE layer. As in their work we use the PerConv tokenisation technique, i.e. given input $x \in \mathbb{N}^{h \times w \times 26}$ we take the output $y \in \mathbb{R}^{h \times w \times 32}$ of $f_c$ and construct $h \times w$ tokens with dimension $d = 32$ that we then feed into the SoftMoE layer. We always use 32 slots and 4 experts for this layer, see (Obando-Ceron et al., 2024) for details on this layer. The resulting embedding is then passed into the two remaining linear layers before being also passed to RNN and used to produce an action or value, equivalent to the description above, compare Figure 9.

Table 6: DR hyperparameters.

| Description | Value |
|---|---|
| $n$ walls to place | Sampled uniformly between $0 - 15$ |
| $n$ onion piles to place | Sampled uniformly between $1 - 2$ |
| $n$ plate piles to place | Sampled uniformly between $1 - 2$ |
| $n$ pots to place | Sampled uniformly between $1 - 2$ |
| $n$ goals to place | Sampled uniformly between $1 - 2$ |

Table 7: PLR specific hyperparameters in addition to the DR hyperparameters.

| Description | Value |
|---|---|
| UED Score | MaxMC (Jiang et al., 2021a) |
| PLR replay probability $\rho$ | 0.5 |
| PLR buffer size | $4,000$ |
| PLR staleness coefficient | 0.3 |
| PLR temperature | 0.1 |
| PLR score ranks | Yes |
| PLR minimum fill ratio | 0.5 |
| $\text{PLR}^{\perp}$ | Yes |
| $\text{PLR}^{\parallel}$ | Yes |
| PLR force unique level | Yes |

Lastly, we describe our networks in terms of parameter count in Table 10.

## A.6 ADDITIONAL ANALYSIS

### A.6.1 EVIDENCE OF OVERFITTING IN OVERCOOKED AGENTS

To verify that agents indeed overfit their training layout in Overcooked we present Table 11 in which we experiment with our weakest performing policy architecture, the CNN-LSTM. This is to be expected but verifying is nonetheless important.

### A.6.2 PERFORMANCE ACROSS LEVELS

To accompany the overall performance measured by reward in the main paper in Table 3 we also measure the mean solved rate on display it in Table 12.

### A.6.3 PERFORMANCE ON INDIVIDUAL LEVELS

We list the performance of every individual method on every single layout in Table 13. Most notable is that some layouts are harder to learn than others. Our agents especially seem to struggle with layouts requiring more complex forms of interaction, i.e. Coordination Ring, Counter Circuit and Forced Coordination. Forced Coordination especially seems difficult to solve as no run achieves noticeable performance on it. This might be due to the specific features of the layout, i.e. agents have access to several objects and need to hand them over the counter to produce any result.

### A.6.4 POPULATION TRAINING DETAILS

To both verify that our implementation is correct and to give an intuition into the performance of the members of the population, we present the training curves over all 8 seeds of training an FCP population in Figure 10. MEP was trained with exactly the same architecture and with the same amount of experience per agent. As in prior work (Zhao et al., 2023) we set the population entropy coefficient during training to $\alpha = 0.01$.

Table 8: ACCEL hyperparameters in addition to the DR hyperparameters.

| Description | Value |
|---|---|
| UED Score | MaxMC (Jiang et al., 2021a) |
| PLR replay probability $\rho$ | 0.8 |
| PLR buffer size | 4,000 |
| PLR staleness coefficient | 0.3 |
| PLR temperature | 0.1 |
| PLR score ranks | Yes |
| PLR minimum fill ratio | 0.5 |
| PLR$^{\perp}$ | Yes |
| PLR$^{\parallel}$ | Yes |
| PLR force unique level | Yes |
| ACCEL Mutation | Overcooked Mutator |
| ACCEL $n$ mutations | 20 |
| ACCEL subsample size | 4 |

Table 9: PAIRED hyperparameters. All PPO hyperparameters are the same between the student and the teacher. The `minimax` implementation follows to original one in (Dennis et al., 2020) and we stick to it too.

| Description | Value |
|---|---|
| $n$ students | 2 |
| UED Score | Relative regret (Dennis et al., 2020) |
| UED first wall sets budget | Yes |
| UED noise dim | 50 |
| PAIRED Creator | OvercookedUED |

### A.6.5 DETAILED RESULTS WITH POPULATIONS

We present detailed zero-shot cooperation results per layout in Tables 14 and 15. As indicated through the averaged performance discussed in the main text, we also find that PAIRED performs best on four of the five individual layouts in terms of zero-shot cooperation.

### A.7 ERROR ANALYSIS NUMBERS

We hypothesise that agents may fail to generalise since observations are hard to generalise from. To test this we design testing layouts that rotate and mirror features of the environment to look for systemic failures, i.e. cases in which an agent does well in one environment but not its mirrored version. While methods generally differed in how well they performed along the same line as previous analysis suggests, the low standard deviations show that any given method performs similarly well on each of the 24 layouts, see Table 16.

### A.7.1 TRAINING CURVES AND EVALUATION

In Figures 11, 12 and 13 we show the returns of our agent during training in seen training levels, as well as the five unseen evaluation levels. The results for the SoftMoE architecture are displayed in Figure 11, the results for the S5 in Figure 12 and the results for the CNN-LSTM in Figure 13. Interestingly, while (SoftMoE) PAIRED performs the best in our evaluations it does not reach the highest training returns, instead it achieves the highest training return while keeping the generalisation gap small.

### A.8 VALIDATING THE IMPLEMENTATION

As an open-source benchmark, we emphasise a correct implementation of the benchmark, including all the baselines. We do so in two important ways. Firstly, we base our implementation on the

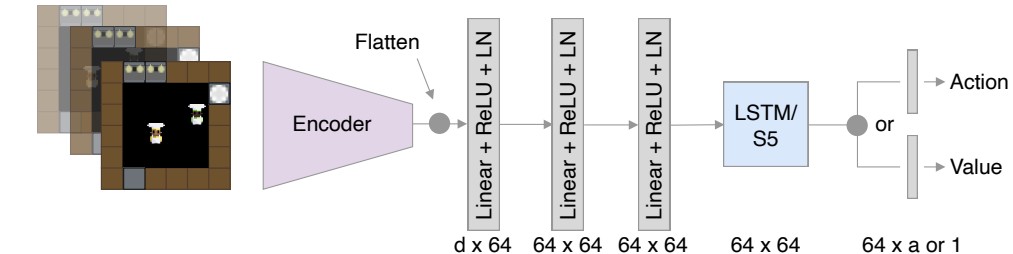

Figure 8: Basic architecture featuring a convolutional encoder and an RNN.

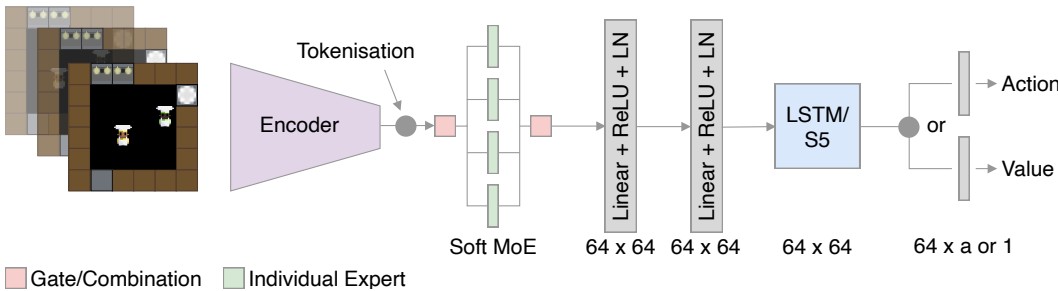

Figure 9: Soft MoE architecture featuring a convolutional encoder, the mixture of experts layer and an RNN.

implementation of the `minimax` benchmark (Jiang et al., 2023), making sure that we use publicly available code for all unsupervised environment design algorithms. Secondly, we test the implementation and adaption of the Overcooked-AI environment by fixing the generated training layouts to a single layout during training. This allows us to train on the 5 classic Overcooked layouts using our implementation. Our implementation is capable of solving these layouts, see Figure 10. We do this in part to argue for the fact that our benchmark is hard to solve and this is not a function of poorly configured or wrongly implemented algorithms.

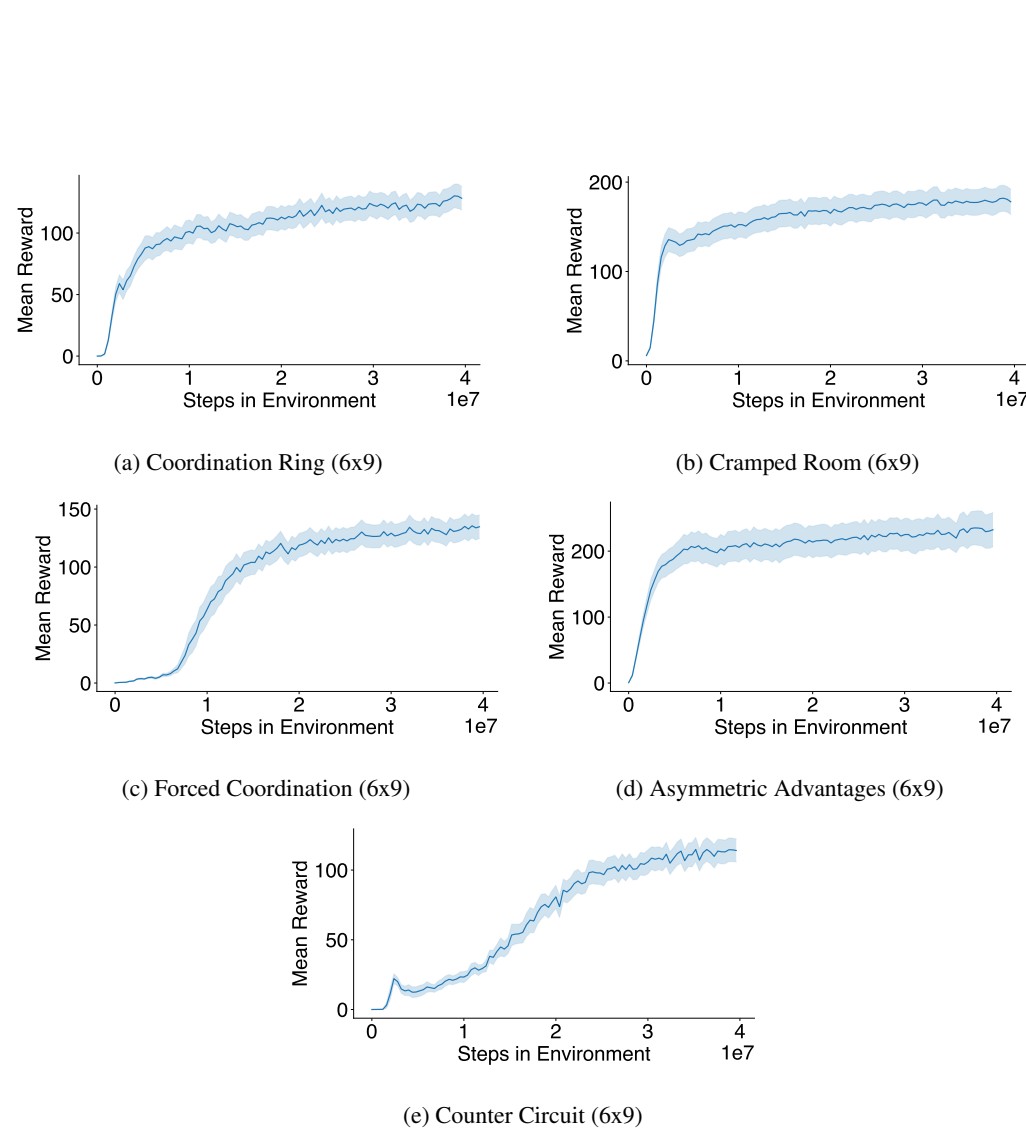

(a) Coordination Ring (6x9)

(b) Cramped Room (6x9)

(c) Forced Coordination (6x9)

(d) Asymmetric Advantages (6x9)

(e) Counter Circuit (6x9)

Figure 10: Runs used for the FCP evaluation populations with random seeds $1 - 8$ for the OGC with bands reporting standard error $\sigma/\sqrt{n}$. Layouts were padded to a total size of 6 x 9 to be compatible with the policies trained via DCD.

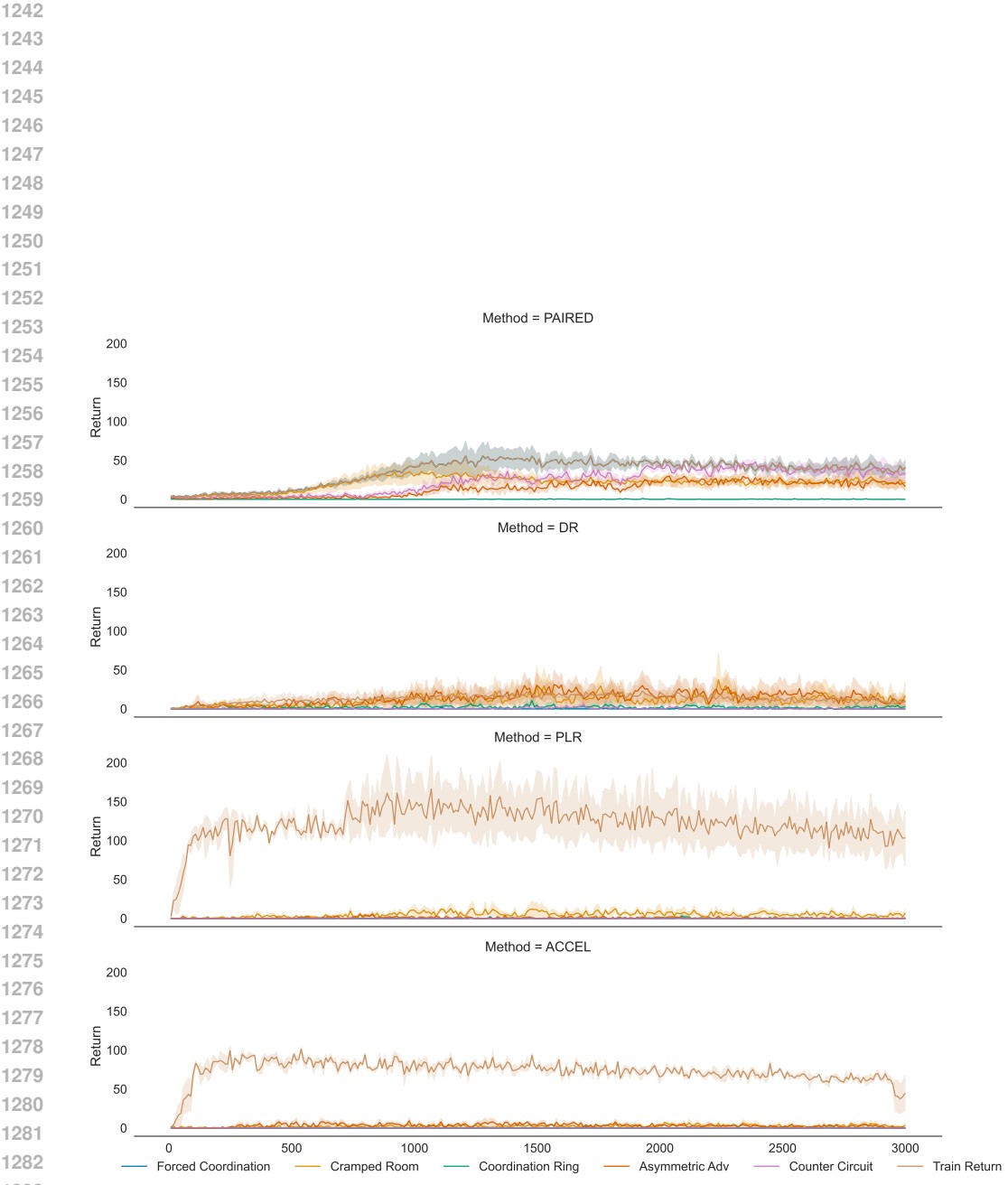

Figure 11: Returns in training and evaluation levels over the duration of training for our **SoftMoE** architecture.

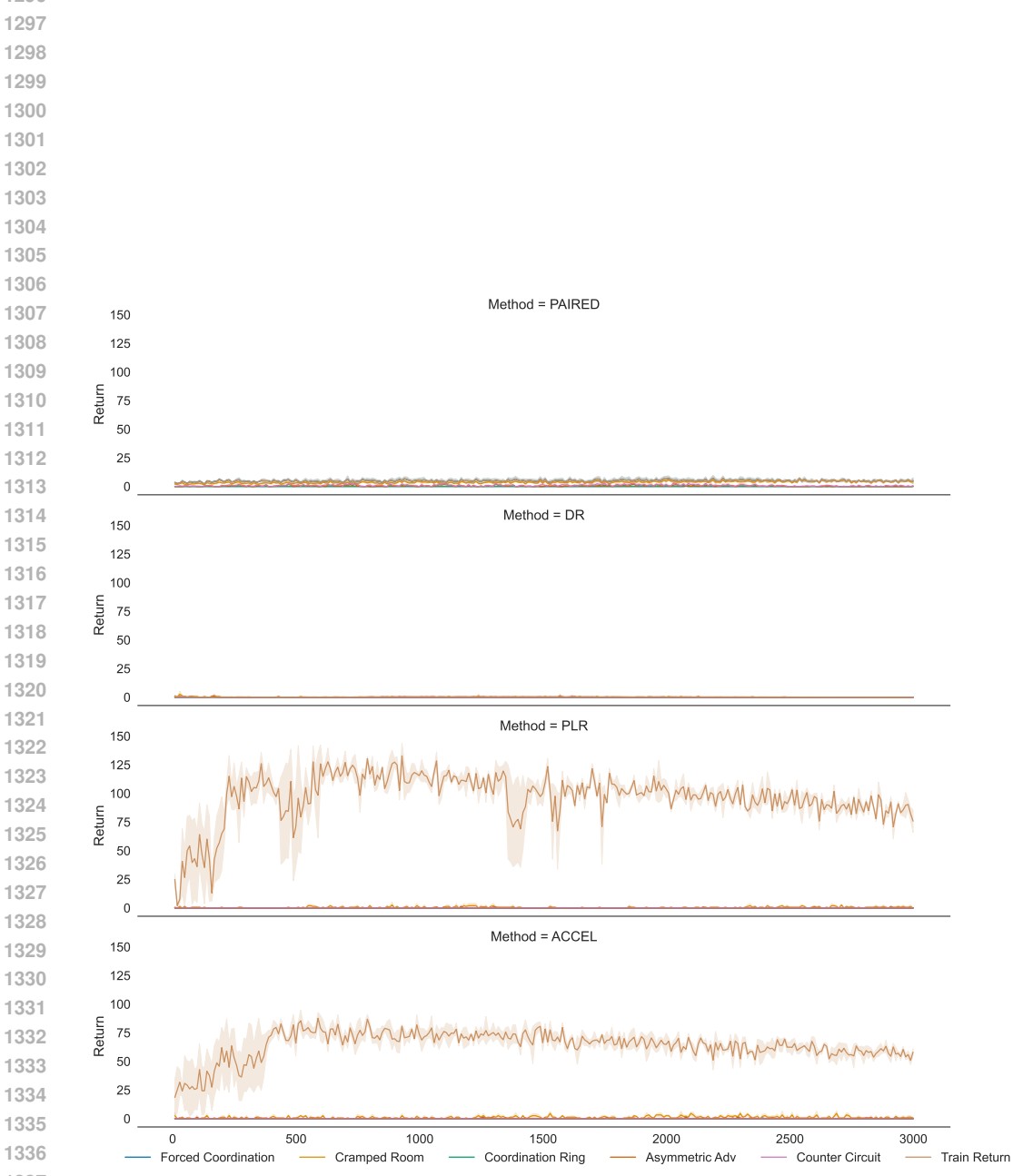

Figure 12: Returns in training and evaluation levels over the duration of training for our **S5** architecture.

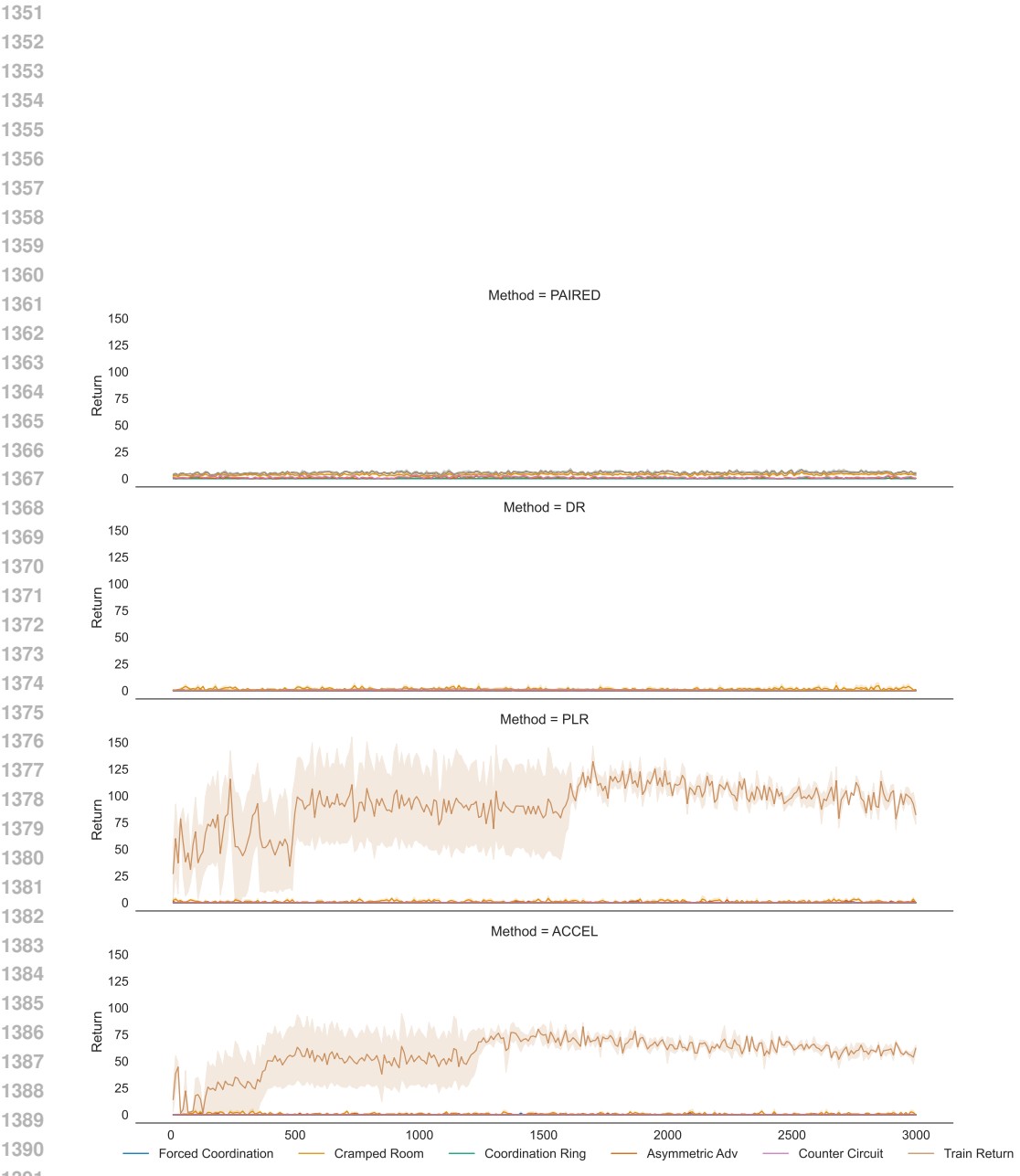

Figure 13: Returns in training and evaluation levels over the duration of training for our **CNN-LSTM** architecture.

Table 10: Number of trainable parameters in each model.

|  | CNN-LSTM | SoftMoE-LSTM | CNN-S5 |
|---|---|---|---|
| Parameter Count | 197,254 | 316,102 | 193,670 |

Table 11: Comparing the layout a CNN-LSTM policy was trained on versus on which it was being evaluated. The policies heavily overfit the training layout. All policies we tested exhibit this property.

|  | Asymm | Cramped | Counter | Forced | Coord |
|---|---|---|---|---|---|
| Asymm | 343.4 | 0.0 | 0.0 | 0.0 | 0.0 |
| Cramped | 1.6 | 185.6 | 0.0 | 0.0 | 0.0 |
| Counter | 0.0 | 0.0 | 128.0 | 0.0 | 0.0 |
| Forced | 0.0 | 0.2 | 0.0 | 141.2 | 0.0 |
| Coord | 0.0 | 0.0 | 0.0 | 0.0 | 144.6 |

Table 12: Mean episode solved rate for the different methods averaged over the respective testing layouts. The best result is shown in **bold**. We report aggregate statistics over three random seeds. As a baseline we include an Oracle version for all architectures, which was trained on the five testing layouts directly.

| Method | CNN-LSTM | SoftMoE-LSTM | CNN-S5 |
|---|---|---|---|
| DR | $0.02 \pm 0.0\%$ | $6.31 \pm 10.1\%$ | $0.00 \pm 0.0\%$ |
| PLR$^{\perp,\parallel}$ | $0.00 \pm 0.0\%$ | $0.33 \pm 0.3\%$ | $0.00 \pm 0.0\%$ |
| Pop. PAIRED | $0.00 \pm 0.0\%$ | $\mathbf{14.62 \pm 7.6}\%$ | $0.00 \pm 0.0\%$ |
| ACCEL$^{\parallel}$ | $0.00 \pm 0.0\%$ | $0.08 \pm 0.1\%$ | $0.00 \pm 0.0\%$ |
| Oracle | $95.40 \pm 7.5\%$ | $99.67 \pm 0.6\%$ | $97.53 \pm 4.1\%$ |

Table 13: Performance on all evaluation layouts. We show the mean episode reward **R** and the mean episode solved rate **SR**. The overall best result per layout is presented in **bold** excluding oracle results.

| Layout | Method | CNN-LSTM | | SoftMoE-LSTM | | CNN-S5 | |
|---|---|---|---|---|---|---|---|
| | | R | SR | R | SR | R | SR |
| Cramped | DR | 1.70 | 0.0% | 1.54 | 0.2% | 0.00 | 0.0% |
| | PLR$^{\perp,\parallel}$ | 1.12 | 0.0% | 5.02 | 2.1% | 0.14 | 0.0% |
| | Pop. PAIRED | 1.44 | 0.0% | **37.02** | **57.7 %** | 0.50 | 0.0% |
| | ACCEL$^{\parallel}$ | 0.92 | 0.0% | 0.60 | 0.0% | 0.60 | 0.0% |
| | Oracle | 241.27 | 96.7% | 245.54 | 100.0% | 189.47 | 99.7% |
| Coord | DR | 0.00 | 0.0% | 0.00 | 0.0% | 0.00 | 0.0% |
| | PLR$^{\perp,\parallel}$ | 0.00 | 0.0% | 0.00 | 0.0% | 0.00 | 0.0% |
| | Pop. PAIRED | 0.00 | 0.0% | **16.78** | **14.6%** | 0.00 | 0.0% |
| | ACCEL$^{\parallel}$ | 0.00 | 0.0% | 0.04 | 0.0% | 0.02 | 0.0% |
| | Oracle | 197.8 | 100.0% | 204.53 | 100.0% | 119.33 | 99.0% |
| Forced | DR | 0.00 | 0.0% | **0.02** | 0.0% | 0.00 | 0.0% |
| | PLR$^{\perp,\parallel}$ | 0.00 | 0.0% | **0.02** | 0.0% | **0.02** | 0.0% |
| | Pop. PAIRED | 0.00 | 0.0% | 0.00 | 0.0% | 0.00 | 0.0% |
| | ACCEL$^{\parallel}$ | 0.00 | 0.0% | 0.00 | 0.0% | 0.00 | 0.0% |
| | Oracle | 196.8 | 100.0% | 204.53 | 100.0% | 133.47 | 94.7% |
| Asymm | DR | 0.58 | 0.1% | 8.64 | 4.4% | 0.00 | 0.0% |
| | PLR$^{\perp,\parallel}$ | 0.08 | 0.0% | 0.10 | 0.0% | 0.08 | 0.0% |
| | Pop. PAIRED | 0.28 | 0.0% | **15.64** | **14.2%** | 0.08 | 0.0% |
| | ACCEL$^{\parallel}$ | 0.14 | 0.0% | 0.04 | 0.0% | 0.02 | 0.0% |
| | Oracle | 220.4 | 100.0% | 277.8 | 98.4% | 247.87 | 99.7% |
| Counter | DR | 0.00 | 0.0% | 0.00 | 0.0% | 0.00 | 0.0% |
| | PLR$^{\perp,\parallel}$ | 0.00 | 0.0% | 0.00 | 0.0% | 0.00 | 0.0% |
| | Pop. PAIRED | 0.00 | 0.0% | **1.38** | 0.0% | 0.00 | 0.0% |
| | ACCEL$^{\parallel}$ | 0.00 | 0.0% | 0.00 | 0.0% | 0.00 | 0.0% |
| | Oracle | 91.2 | 77.3% | 152.73 | 100.0% | 84.93 | 94.7% |

Table 14: Zero-shot results using SoftMoE-LSTM policies playing with an FCP and MEP population of experts trained on the respective layout exclusively. We report the *mean episode reward* and standard deviation. The best result per layout is put in **bold**.

| Method | Asymm | Counter | Cramped | Forced | Coord |
|---|---|---|---|---|---|
| | | | FCP | | |
| Random | $7.43 \pm 12.19$ | $8.89 \pm 4.65$ | $66.02 \pm 38.28$ | $1.95 \pm 1.92$ | $20.49 \pm 7.82$ |
| Stay | $5.32 \pm 12.07$ | $0.38 \pm 1.11$ | $20.67 \pm 33.05$ | $0.00 \pm 0.00$ | $0.95 \pm 2.73$ |
| Oracle | $126.44 \pm 27.13$ | $22.63 \pm 7.82$ | $120.9 \pm 10.86$ | $22.08 \pm 12.89$ | $59.64 \pm 22.17$ |
| DR | $18.18 \pm 1.69$ | $6.86 \pm 5.27$ | $65.05 \pm 5.15$ | $1.09 \pm 0.21$ | $17.88 \pm 10.27$ |
| $PLR^{\perp,\parallel}$ | $7.64 \pm 0.89$ | $5.60 \pm 1.29$ | $60.35 \pm 6.89$ | $1.76 \pm 0.86$ | $21.90 \pm 1.26$ |
| Pop. PAIRED | $\mathbf{24.51 \pm 3.44}$ | $\mathbf{11.11 \pm 1.67}$ | $\mathbf{81.92 \pm 6.33}$ | $1.59 \pm 0.57$ | $\mathbf{29.72 \pm 4.72}$ |
| $ACCEL^{\parallel}$ | $8.60 \pm 0.98$ | $10.23 \pm 0.85$ | $65.46 \pm 4.62$ | $\mathbf{1.81 \pm 1.25}$ | $19.19 \pm 1.93$ |
| | | | MEP | | |
| Random | $8.0 \pm 9.12$ | $22.46 \pm 13.34$ | $58.33 \pm 34.83$ | $2.55 \pm 2.76$ | $31.85 \pm 19.69$ |
| Stay | $4.86 \pm 7.21$ | $5.2 \pm 10.85$ | $31.55 \pm 47.13$ | $0.0 \pm 0.0$ | $1.53 \pm 3.61$ |
| Oracle | $135.07 \pm 30.27$ | $39.33 \pm 13.53$ | $138.07 \pm 10.0$ | $56.1 \pm 25.41$ | $67.86 \pm 10.89$ |
| DR | $19.32 \pm 0.39$ | $18.04 \pm 5.75$ | $62.77 \pm 7.22$ | $1.69 \pm 0.67$ | $30.35 \pm 4.42$ |
| $PLR^{\perp,\parallel}$ | $7.53 \pm 0.92$ | $21.23 \pm 1.91$ | $57.2 \pm 4.4$ | $2.45 \pm 1.23$ | $2.45 \pm 1.23$ |
| Pop. PAIRED | $\mathbf{24.33 \pm 2.27}$ | $\mathbf{23.72 \pm 4.0}$ | $\mathbf{82.23 \pm 9.38}$ | $\mathbf{2.96 \pm 1.56}$ | $\mathbf{37.1 \pm 6.28}$ |
| $ACCEL^{\parallel}$ | $9.3 \pm 0.71$ | $18.33 \pm 1.96$ | $56.72 \pm 4.15$ | $2.21 \pm 1.57$ | $28.52 \pm 1.55$ |

Table 15: Zero-shot results using SoftMoE-LSTM policies playing with an FCP and MEP population of experts trained on the respective layout exclusively. We report the *mean solved rate* and standard deviation. The best result per layout is put in **bold**.

| Method | Asymm | Counter | Cramped | Forced | Coord |
|---|---|---|---|---|---|
| Random | $8.52 \pm 17.52\%$ | $5.00 \pm 6.70\%$ | $69.43 \pm 38.45\%$ | $0.00 \pm 0.00\%$ | $30.89 \pm 3.83\%$ |
| Stay | $6.81 \pm 18.04\%$ | $0.02 \pm 0.14\%$ | $21.75 \pm 33.71\%$ | $0.00 \pm 0.00\%$ | $0.14 \pm 0.74\%$ |
| Oracle | $69.67 \pm 16.39\%$ | $27.39 \pm 19.02\%$ | $31.30 \pm 20.97\%$ | $92.02 \pm 1.19\%$ | $96.96 \pm 2.23\%$ |
| DR | $24.19 \pm 4.60\%$ | $4.56 \pm 5.32\%$ | $72.11 \pm 6.29\%$ | $0.01 \pm 0.01\%$ | $23.76 \pm 18.85\%$ |
| $PLR^{\perp,\parallel}$ | $8.84 \pm 1.31\%$ | $2.04 \pm 0.95\%$ | $68.14 \pm 1.21\%$ | $\mathbf{0.11 \pm 0.12}\%$ | $30.89 \pm 3.83\%$ |
| Pop. PAIRED | $\mathbf{32.48 \pm 4.00}\%$ | $\mathbf{7.91 \pm 1.38}\%$ | $\mathbf{85.54 \pm 6.08}\%$ | $0.09 \pm 0.07\%$ | $\mathbf{48.31 \pm 11.08}\%$ |
| $ACCEL^{\parallel}$ | $9.58 \pm 1.12\%$ | $6.79 \pm 0.91\%$ | $69.01 \pm 2.03\%$ | $0.06 \pm 0.06\%$ | $24.13 \pm 6.01\%$ |
| | | | MEP | | |
| Random | $9.25 \pm 2.02\%$ | $36.04 \pm 4.38\%$ | $67.75 \pm 5.48\%$ | $0.00 \pm 0.00\%$ | $54.9 \pm 5.55\%$ |
| Stay | $4.91 \pm 1.46\%$ | $5.85 \pm 2.71\%$ | $29.56 \pm 5.92\%$ | $0.00 \pm 0.00\%$ | $1.02 \pm 0.51\%$ |
| Oracle | $91.02 \pm 1.12\%$ | $52.60 \pm 11.37\%$ | $96.86 \pm 2.27\%$ | $56.16 \pm 21.85\%$ | $75.23 \pm 0.91\%$ |
| DR | $26.34 \pm 3.55\%$ | $27.41 \pm 10.31\%$ | $70.78 \pm 4.23\%$ | $0.05 \pm 0.07\%$ | $50.07 \pm 6.67\%$ |
| $PLR^{\perp,\parallel}$ | $8.24 \pm 1.28\%$ | $33.76 \pm 4.89\%$ | $65.38 \pm 4.55\%$ | $0.28 \pm 0.41\%$ | $50.97 \pm 4.01\%$ |
| Pop. PAIRED | $\mathbf{32.79 \pm 1.81}\%$ | $\mathbf{36.48 \pm 8.14}\%$ | $\mathbf{80.60 \pm 6.74}\%$ | $\mathbf{0.38 \pm 0.51}\%$ | $\mathbf{55.23 \pm 8.42}\%$ |
| $ACCEL^{\parallel}$ | $10.10 \pm 0.11\%$ | $25.94 \pm 4.31\%$ | $66.52 \pm 3.34\%$ | $0.18 \pm 0.16\%$ | $48.80 \pm 2.21\%$ |

| Method | SoftMoE-LSTM | CNN-S5 | CNN-LSTM |
|---|---|---|---|
| DR | $11.91 \pm 0.8$ | $0.00 \pm 0.0$ | $1.96 \pm 0.3$ |
| $PLR^{\perp,\parallel}$ | $4.39 \pm 0.4$ | $0.49 \pm 0.2$ | $0.65 \pm 0.2$ |
| Pop. PAIRED | $36.83 \pm 1.1$ | $0.58 \pm 0.2$ | $0.59 \pm 0.1$ |
| $ACCEL^{\parallel}$ | $2.91 \pm 0.4$ | $1.69 \pm 0.3$ | $0.78 \pm 0.2$ |

Table 16: Performance on mirrored and rotated levels, illustrated in Figure 6.

