# OpenReview forum: "The Overcooked Generalisation Challenge"
_ICLR.cc/2025/Conference — Submitted to ICLR 2025_

### Official Review · Reviewer_PTdY · 2024-10-20

**Soundness:** 3
**Presentation:** 3
**Contribution:** 3
**Rating:** 6
**Confidence:** 3

**Summary:**

This paper introduces the Overcooked Generalisation Challenge (OGC), which is a benchmark for testing RL agents' ability to cooperate with new partners in unseen environments. It focuses on zero-shot coordination to improve human-AI collaboration in complex scenarios. The study shows that current algorithms struggle with this challenge, highlighting the need for more adaptable cooperative agents.

**Strengths:**

- Originality: The paper introduces a new benchmark, the Overcooked Generalisation Challenge (OGC), which evaluates RL agents' zero-shot cooperation abilities with new partners in previously unseen layouts, addressing gaps in existing cooperative AI research.

- Quality: The study provides extensive empirical results, and includes detailed error analysis, showcasing both successful and challenging areas for current algorithms.

- Clarity: The paper is well-structured, clearly presenting the motivations, methodology, and findings. It is easy for readers to follow the research and apply its insights.

- Significance: The OGC fills the gap in cooperative AI research by focusing on zero-shot coordination, which is essential for real-world applications of AI in dynamic and unpredictable environments.

**Weaknesses:**

- The challenge restricts layout sizes to ensure observability and compatibility with CNN-based encoders, which may limit the exploration of more natural and scalable representations for complex environments.

- The OGC environment, with its emphasis on running parallel layouts and complex training, may demand substantial computational resources. This could limit accessibility for researchers without high-performance computing capabilities and restrict broader experimentation and adoption of the benchmark.

**Questions:**

How do you plan to address the constraints on layout size in future iterations of the OGC to explore more scalable or natural scene representations? Would methods like graph-based representations or item embeddings be a viable approach?

What are the specific computational requirements for running the OGC, and have you established any benchmarks or guidelines to help researchers estimate the necessary resources for effective experimentation?

---

### Official Review · Reviewer_3m6E · 2024-10-31

**Soundness:** 3
**Presentation:** 3
**Contribution:** 3
**Rating:** 6
**Confidence:** 4

**Summary:**

This works proposes OvercookedUED, an GPU-based cooperative multi-agent environment that is compatible with UED methods. The proposed environment is more complex than existing UED-compatible environments. It also aims to connect UED with ad-hoc teamwork literature by allowing agents to be tested with unseen partners under unseen game levels. Empirically, existing UED methods struggle to produce strong agents in OvercookedUED. Further, UED-produced agents tend to cooperate better when they are more generally capable.

**Strengths:**

- The paper is well written and easy to understand.
- I really like that the authors aim to connect the UED and ad-hoc teamwork literature through this benchmark. The paper highlights clearly what is the gap of existing UED-compatible environments, i.e., environment and partner generalization, and thus is well motivated and novel.
- The evaluation seems to be thorough with many baselines and evaluation scenarios
- The project will be open-source

**Weaknesses:**

Though I really like the paper at a high-level, there are quite a few significant improvements that can make the paper more mature:

- The environment assumes a single recipe in the game and thus missing an important aspect of the game. There are variants of Overcooked that use multiple recipes that make the game more interesting and require more complex coordination [1,2,3]. I believe including more possible recipes improves the depth of the environment significantly, arguably adding another important layer of generalization.
- line 288: the observation space is not explained in details, it is especially hard for readers without prior Overcooked knowledge.
- It is not clear if the UED agents were just undertrained or the UED algorithms themselves struggle to generate complex layout. I wish the authors include training curves showing the performance on seen and unseen layouts of UED agents.
- The result of per layout generalization performance is missing from the main paper. I believe Table 13 shows this result but is not included in the main paper. I compare Table 13 and Table 14 (which is the raw results of Fig 5) and see that the self-play performance is actually lower than *test cross-play* performance, hinting that the agents are being *carried* by the FCP test partners. I believe that this observation is important and should be included and discussed in the main text.
- FCP does not explicitly generate diverse agents, which might not be a robust method for generating test partners.


Minor comments:
- Random and Stay baselines (Fig 5) are never introduced
- Fig 4 sentence 2: I find the sentence to not be consistent with the figure. Some of the generated layouts are definitely solvable.
- line 54: empty space on the right
- last line of page 2: the citation should not have parentheses as it is a part of a sentence


[1] Wu, Sarah A., et al. "Too many cooks: Bayesian inference for coordinating multi‐agent collaboration." Topics in Cognitive Science 13.2 (2021): 414-432.

[2] Yu, Chao, et al. "Learning Zero-Shot Cooperation with Humans, Assuming Humans Are Biased." The Eleventh International Conference on Learning Representations.

[3] Charakorn, Rujikorn, Poramate Manoonpong, and Nat Dilokthanakul. "Generating diverse cooperative agents by learning incompatible policies." The Eleventh International Conference on Learning Representations. 2023.

**Questions:**

- Is it possible to **only** test partner generalization under the proposed environment (e.g., something similar to Fig. 5 but under seen layouts)? This would allow user to isolate between two types of generalization, diagnose, and develop better UED+ZSC algorithms.

---

### Official Review · Reviewer_Kg98 · 2024-11-01

**Soundness:** 1
**Presentation:** 2
**Contribution:** 1
**Rating:** 3
**Confidence:** 4

**Summary:**

This paper connects the Overcooked-AI implementation from `JaxMARL` [1] with the `minimax` [2] library for Unsupervised Environment Design (UED) to create a new benchmark, the Overcooked Generalization Challenge (OGC).

OGC is meant to assess the generalization and coordination ability of agents when faced with novel partners in levels unseen during train time, where each level is a specific instantiation of the overarching Dec-UPOMDP.

The authors then evaluate four popular UED methods combined with three different policy architectures by looking at the mean episode return in self-play on held-out layouts (Table 3) and mean episode return when paired with fictitious co-play (FCP) policies in held-out layouts (Figure 5). In both cases, the held-out layouts are the 5 original layouts introduced in Overcooked-AI [3].

Finally, authors perform some analysis in an attempt to explain the low performance of DCD methods in the benchmark.

[1] https://github.com/FLAIROx/JaxMARL/tree/main/jaxmarl/environments/overcooked

[2] https://github.com/facebookresearch/minimax

[3] https://github.com/HumanCompatibleAI/overcooked_ai

**Strengths:**

As the authors have pointed out, the vast majority of the coordination literature studies the coordination challenges that arise when paired with novel partners in the same environment as seen as train time (though with possible variations, such as different deck orderings in Hanabi [1].)

As a result, there is indeed a significant gap in the coordination literature regarding test-time generalization to scenarios unseen during training, including to new layouts, different number of agents, new dynamics or new reward functions. A benchmark that allows to jointly evaluate the coordination and generalization ability of AI agents is a welcome contribution, and could drive significant algorithmic innovations. However, for the reasons explained below, I do not believe this paper fulfills this role.

Other strengths include the writing, which I found clear and easy to follow, the commitment to open-sourcing the benchmark, and the thorough reporting of hyperparameters in the appendix. Finally, adopting the Dec-UPOMDP formalism allows to more rigorously define the extent of generalization the model should reach.

[1] The Hanabi Challenge (https://arxiv.org/abs/1902.00506)

**Weaknesses:**

**1. Limited Scope**

While authors adopt the Dec-UPOMDP formalism, their proposed challenge only evaluates generalization to new layouts. This means they only assess generalization to new states, but not to new observation functions, reward functions or dynamics. As such, the whole benchmark is in fact simpler than the formalism suggests, and is equivalent to simply augmenting the set of initial states in Overcooked.

This problem is exacerbated by the environment being fully observable and deterministic, which in many layouts means that agents can simply react to what their partner is doing for a good, if not optimal, policy. See Section 7 of [1] for a discussion of partial observability and what that implies for limitations of Overcooked as a coordination challenge.

**2. Unclear Contributions**

The authors claim to introduce a new environment, OvercookedUED, and it is listed as one of their 3 main contributions of the paper on page 2. That said, from the description in section 5.1, OvercookedUED is simply a wrapper of the JaxMARL implementation of Overcooked-AI (i.e. Overcooked-JAX) that connects it to the `minimax` library. Since Overcooked-JAX already comes with a way of easily creating new layouts by providing an ASCII string [2], OvercookedUED appears to be only a minor contribution.

Relatedly, Table 2 seems to report the steps-per-second achieved by running Overcooked-JAX, and it is unclear what information it carries that is specific to the authors' contribution.

**3. Different design space for different UED methods**

The description of OvercookedUED in Section 5.1 lacks key details on how the layout generation is performed for each UED method. Furthermore, there appear to be significant differences in the design space for each method, which can in turn affect performance. Based on the paper:

- For PAIRED, new layouts are generated by the teacher placing objects "sequentially and in a deterministic order". Since the teacher conditions only on the unfinished layout, does that mean that the number of objects of each type is fixed for all layouts generated by PAIRED?

- PLR can select 1 or 2 piles of onions, bowls, pots and serving locations, so the number of objects of each type is clearly variable. The authors do not specify the sampling procedure for the walls, which can greatly influence the layout.

- ACCEL can remove/add walls and move all other objects in the layout, but *not* remove them. There's also no mention on whether it can move agent starting positions.

Because the design space for each method can greatly influence its performance, the authors should clearly detail how each method operates, and also eliminate any discrepancy. In other words, *all layouts theoretically obtainable by one method should also be obtainable by the other ones*. The descriptions in the paper indicate that this is not the case.

**4. No assessment of policy learning ability**

The authors evaluate three different policy architectures. In Table 11, they show that one unspecified architecture can learn to overfit to a single layout. However, they provide no evidence that any of those architectures can learn a policy over multiple layouts. At a minimum, the authors should train each of the architectures on the 5 held-out layouts and show that they can indeed learn a general policy. They could then evaluate the performance gap between those expert agents and those trained in UED.

Without that experiment, it is impossible to conclude whether the poor performance in Section 6 is due to the UED methods failing to provide a suitable curriculum or to the policy networks being unable to learn a general policy.

**5. No assessment of robustness of FCP policies**

Coordination is evaluated in the ad-hoc teamplay [3] framework (despite the authors never using that term and failing to cite Stone et al. 2010), where the final policy is paired with a pool of test-time partners. In the case of the paper, the pool is obtained by Fictitious Co-play (FCP), but the authors do not evaluate whether those policies are themselves robust to new partners. This poses two problems.

First, this evaluation pool is limited and likely not very diverse. Authors claim that "*There is currently no algorithm to train a population of diverse agents over a distribution of levels.*", but given that each level simply corresponds to a different initial state, algorithms such as TrajeDi [4] or ComeDi [5] could apply out of the box.

Second, there is no evaluation in the paper showing that FCP policies are robust to unseen partners. As such, poor returns in cross-play could be the fault of the trained policy or of the FCP policies themselves. See [6] for an extreme example of this.

**6. Other minor issues**

- The authors misuse "Zero-Shot Coordination"; it is a specific setting introduced in [7] and requiring that test time partners be trained with the same algorithm used to train the policy being evaluated.
- The authors misuse "Other-play"; it is a method introduced in [7], not an evaluation setting. Instead, the authors should use the term "cross-play".
- "*Notice that the training levels in which our model performs well in are similar to Asymmetric Advantages and Cramped Room, while the worst levels are similar to the other 3 evaluation levels*". This statement is not substantiated by Figure 7.
- Most figures use the visuals from Overcooked-AI but figure 7 uses the visuals form OvercookedJAX, without explanation. This can be confusing to authors which are not familiar with both works.

[1] Who Needs to Know?, https://arxiv.org/pdf/2306.09309

[2] Overcooked-JAX, https://github.com/FLAIROx/JaxMARL/tree/main/jaxmarl/environments/overcooked

[3] Stone et al., https://ojs.aaai.org/index.php/AAAI/article/view/7529

[4] TrajeDi, https://proceedings.mlr.press/v139/lupu21a/lupu21a.pdf

[5] ComeDi, https://arxiv.org/pdf/2310.15414

[6] ADVERSITY, https://openreview.net/pdf?id=uLE3WF3-H_5

[7] Other-Play, https://arxiv.org/abs/2003.02979

**Questions:**

1. Can the authors please clarify the sampling procedure for each of the UED methods?
2. Can authors clarify what OvercookedUED provides on top of OvercookedJAX and minimax, from an implementation perspective?

---

### Official Review · Reviewer_QU9G · 2024-11-02

**Soundness:** 3
**Presentation:** 3
**Contribution:** 3
**Rating:** 8
**Confidence:** 4

**Summary:**

MARL research has focused on studying how to generalize across novel partners on the same task. However, there is little work on generalizing to novel partners across novel tasks in the Overcooked-AI environment. This work introduces a new challenge for using DCD methods on fully cooperative tasks in Overcooked. They show how SOTA DCD methods cannot build agents that learn to coordinate with novel partners across novel problems, even with recent advances in neural networks, making this an open challenge for the MARL community.

**Strengths:**

This paper was a compelling first stab at building agents capable of generalizing across partners and problems. The authors did their due diligence with showing the speed of their environment, the performance of difference SOTA algorithms for UED across a variety of network architectures, and justifying the distinction between this work and prior methods for enabling ZSC, such as population based approaches. Overall, the implementation and evaluation details seem well thought through and justified, and the results point towards a clear gap in the multi-agent UED literature.

**Weaknesses:**

While most of the paper was well written, Section 6 on benchmarking the challenge was a little confusing. Specifically, the difference between generalization performance and zero-shot cooperation performance could've been written with more clarity. It was not obvious if the authors were referring to generalization with partners during training across novel layouts, generalization to partners not seen during training for a single layout within distribution of the generator, and generalization in to novel partners across novel layouts.

Also, Figure 7 is a little unclear. While it is explained in the error analysis section, for someone quickly looking at it it is not obvious what the takeaway should be, or how the number of visits should relate to the return. (as a nitpick, the colors for the number of visits could be in a different pallet to be more obvious, not super necessary but may be helpful). In general, while benchmarking and understanding the limits of the challenges of existing UED approaches is very important and will lead to a high quality paper, the methods for analysis could use more than a few sentences of justification if possible. I.e. if the authors think state-coverage is a useful metric for ability to cooperate and that is why they choose to analyze the frequency of cells visited, then this motivation should be spelled out.

**Questions:**

For the error analysis, it is interesting that the models perform well in layouts like asymmetric advantages and cramped room. Both of those layouts are capable of being solved by a single agent to get a high cross-play reward. Can you draw any conclusions about how existing UED objectives might be biasing agents towards solving single agent problems even though the reward scheme technically makes it cooperative?

---

### Meta-Review · Area_Chair_ZsFP · 2024-12-19

**Metareview:**

The paper focuses on an interesting problem and has great intentions, however, it has significant limitations as highlighted by reviewer Kg98. Notably, it is a wrapper around two existing libraries, has only a constrained setting to test generalization and uses a different design space for different UED algorithms. We want to point out one detail that was missing in the discussion, the ACCEL paper intentionally randomized the number of total walls for DR and PLR to make the comparison with ACCEL fair, since otherwise this is a confounder. While it wasn't done perfectly, it is clearly the case that fixing the number biases the result to one of the methods. I would suggest re-running the experiments with this setup and adding more diverse novel tasks to get something a bit more robust.

**Additional Comments On Reviewer Discussion:**

There was an extensive discussion with reviewer Kg98, focused on the comparison between different methods with different UED design spaces. I am inclined to side with the reviewer that this should have been handled better for a fairer comparison.

---

### Decision · Program_Chairs · 2025-01-22

Reject